# Application of Lactic Acid Bacteria (LAB) in Sustainable Agriculture: Advantages and Limitations

**DOI:** 10.3390/ijms23147784

**Published:** 2022-07-14

**Authors:** Jegadeesh Raman, Jeong-Seon Kim, Kyeong Rok Choi, Hyunmin Eun, Dongsoo Yang, Young-Joon Ko, Soo-Jin Kim

**Affiliations:** 1Agricultural Microbiology Division, National Institute of Agricultural Science, Rural Development Administration, Wanju-Gun 55365, Jeollabuk-do, Korea; jegadeesh_ooty@rediffmail.com (J.R.); jjsskk@korea.kr (J.-S.K.); kyjggang12@korea.kr (Y.-J.K.); 2Metabolic and Biomolecular Engineering National Research Laboratory, Korea Advanced Institute of Science and Technology (KAIST), Daejeon 34141, Korea; choikr@kaist.ac.kr (K.R.C.); hmeun2004@kaist.ac.kr (H.E.); dosoyang@kaist.ac.kr (D.Y.)

**Keywords:** lactic acid bacteria, sustainable, agricultural, plant growth, biocontrol, bioremediation

## Abstract

Lactic acid bacteria (LAB) are significant groups of probiotic organisms in fermented food and are generally considered safe. LAB regulate soil organic matter and the biochemical cycle, detoxify hazardous chemicals, and enhance plant health. They are found in decomposing plants, traditional fermented milk products, and normal human gastrointestinal and vaginal flora. Exploring LAB identified in unknown niches may lead to isolating unique species. However, their classification is quite complex, and they are adapted to high sugar concentrations and acidic environments. LAB strains are considered promising candidates for sustainable agriculture, and they promote soil health and fertility. Therefore, they have received much attention regarding sustainable agriculture. LAB metabolites promote plant growth and stimulate shoot and root growth. As fertilizers, LAB can promote biodegradation, accelerate the soil organic content, and produce organic acid and bacteriocin metabolites. However, LAB show an antagonistic effect against phytopathogens, inhibiting fungal and bacterial populations in the rhizosphere and phyllosphere. Several studies have proposed the LAB bioremediation efficiency and detoxification of heavy metals and mycotoxins. However, LAB genetic manipulation and metabolic engineered tools provide efficient cell factories tailor-made to produce beneficial industrial and agro-products. This review discusses lactic acid bacteria advantages and limitations in sustainable agricultural development.

## 1. Introduction

Agriculture is an important economic sector in many countries, and according to the FAO, 37% of the global land area is dedicated to agriculture [1]. Conventional farming uses chemical fertilizers and pesticides to boost yield and production. However, increasing the usage of chemical fertilizers affects ecological balance and food safety and is the main causative factor of land and water pollution. In recent years, sustainable agriculture has drawn the attention of the global community, and this approach promotes organic farming in the context of soil health, securing environmental quality. The interaction between plants and microbes is an integral part of sustainable agriculture. Therefore, microbial-based agricultural practices and advancements could promote plant health and soil fertility. Indeed, this approach may secure food for people and ensure a profit and global health. Agricultural microbiology deals with the plant-associated microbes and their application to minimize disease and increase soil fertility. In addition, soil fertility is improved by the microbes’ decomposition process and the addition of adequate plant nutrients. The interaction between plants and beneficial microorganisms in the rhizosphere is a symbiotic relationship: both species are benefited. In addition, the microbes play a crucial role in plant growth promotion, improving nutrient acquisition, and protecting the plant from biotic and abiotic stress [2,3]. The genera *Rhizobium*, *Bacillus*, and *Pseudomonas*, as well as mycorrhizal fungi, are beneficial microorganisms in the soil [4]. In contrast, several pathogenic fungi and bacterial species seriously affect the yield and quality of agricultural products. Therefore, plant pathogenic fungi and insects are enormous challenges to sustainable agriculture. For this reason, developing highly potential and novel antimicrobial agents is a high priority to increase the yield and raise incomes for farmers. LAB are ubiquitous members of many plant microbiomes, but functional information regarding the interaction between LAB and their hosts is lacking. In addition, plant-root-associated rhizobacteria are abundant in soil, while LAB are minimal and not dominant in organic farming soil [5]. LAB promote seed germination, increase soil fertility, aeration, and solubility, alleviate various abiotic stress, and neutralize toxic gasses. However, LAB plant-growth-promoting properties are not well explored and have limited evidence in the literature.

A comprehensive understanding of LAB is that they are a phylogenetically diverse group of Gram-positive bacteria. They are rod-shaped or spherical, non-spore-forming, and catalase-negative bacteria. LAB strains are fastidious microbes, require expensive media nitrogen sources, and have limited biosynthetic pathways. LAB have GRAS (Generally Recognized as Safe) status by the Food and Drug Administration. They are safe for human and animal consumption and have become ideal for commercial development [6,7]. LAB strains show probiotic properties and are used in the food and dairy industry. Among them, Lactobacilli and cocci have been predominantly used in food industry. Lactobacillus species transform undesirable flavor substances in the environment. At the same time, they are decomposing macromolecules and complex biomolecule substances. LAB produce short-chain fatty acids, amines, organic acids, bacteriocins, vitamins, and exopolysaccharides [8]. Bacteriocin metabolites are toxic to microbes and are the most promising for developing antibiotic drugs with probiotic properties. In addition, organic acids are the prominent secondary metabolites that exhibit antifungal activity and preservative effects in fermented food and silage [9]. However, most inhibitory compounds are secondary metabolites produced after 48 h of LAB fermentation [10]. Furthermore, LAB fatty acid metabolites exhibit antimicrobial properties and protect host cells against infections [11]. LAB-derived unsaturated fatty acids and hydroxyl unsaturated fatty acids exhibit antifungal activity. Furthermore, glycolipid biosurfactants play a significant role in preventing bacterial attachment and eradicating biofilm [12]. In addition, biosurfactants have broad applications in bioremediation, biodegradation, and the agricultural, cosmetic, and pharmaceutical industries. LAB metabolites indicated a synergistic effect in pathogenic microbes [13]. Hashemi and Jafarpour demonstrated that LAB-incorporated Konjac-based edible film prevents fungal growth in fresh fruits and positively impacts their shelf life [14]. Furthermore, several studies have shown that LAB could produce antifungal and antibacterial substances to inhibit the growth of pathogenic microbes [7]. In addition, LAB culture conditions such as temperature, low pH, and anaerobic conditions inhibit various mold and food-borne pathogens [8]. Thus, the LAB characterized by antagonistic properties are crucial to countering potential pathogens [15]. LAB strains are a promising biocontrol agent; they have a plant growth stimulation effect and inhibit phytopathogenic microbes [3]. In addition, LAB controls the insects and pests and is involved in bioremediation, and the general agricultural application of LAB is illustrated in Figure 1.

## 2. Lactic Acid Bacteria (LAB)

LAB play a multifaceted role in the food, agricultural, and medicine sectors and has GRAS (Generally Recognized as Safe) status by the Food and Drug Administration [16]. They are safe for human and animal consumption and have become ideal for commercial development [6,7]. LAB species are used in many food and feed industries, and those industries are constantly seeking potential strains to enhance sensor and product quality. They are isolated from decomposing plant material, vegetables, fruits, dairy products, fermented food, fermented beverages, silages, juices, sewage, and the gastrointestinal tracts and cavities of humans and animals [17,18,19,20] (Figure 2). Although LAB identification is challenging, contemporary 16S rDNA sequencing techniques accurately identify individual strains, but phenotypic methods are unreliable [21]. Therefore, the molecular taxonomy and genome sequencing of LAB strains become an effective method for identifying species levels. *Lactobacillus plantarum*, *L. casei*, *Lactococcus*, *Bifidobacterium*, and *Streptococcus lactis* are isolated from the intestinal tract of animals and fermented food [22]. *L. acetotolerans*, *L. pontis*, and *L. suebicus* species show high survival rates in the cow gut [23]. LAB constitute part of the animal gut, and fermented food and silage are recognized as the primary niche of LAB activity. They have been clustered into two different groups, homo- and hetero-fermentative strains, based on lactic acid (LA) yield. Homo-fermentation yields two molecules of LA, while hetero-fermentation yields one molecule of LA and one molecule of ethanol or acetic acid by utilizing glucose. Homo-fermentative strains are commercially important, and they can produce optically pure LA by downstream processes [24]. Lactic acid (LA) is a by-product of metabolic activities produced by LAB. Therefore, silage can be considered a primary source to transmit and deliver the probiotic LAB species. Fermented cattle milk is an LA source that enhances food quality and flavor.

LAB are widespread in dairy and agro-product development, utilizing carbon as an energy source. LAB-based agro-products are safe, eco-friendly, have low production costs, and have fast development rates. Most plant-growth-promoting microorganisms (PGPM) are bacteria/fungi that can promote plant growth, suppress pathogenicity, and accelerate nutrient availability and uptake. For some time, LAB have been used in agriculture as biofertilizers and biocontrol agents to promote plant growth, but the mechanisms of LAB have yet to be explored. LAB are diversified in the phyllosphere, the endosphere in the seed, and the rhizosphere of many plants [3,25]. Several LAB strains were isolated from rhizospheres. In addition, *L. lactis* species have been isolated from horticultural and fruit crop plantations [26,27]. They facilitate tissue repair in damaged plants, while cellular components are released for defense/interaction. In the rhizosphere, plants release various chemical substances, including 20–40% of the carbohydrates and organic acids [28]. Those metabolites attract the LAB and colonize the root systems’ surface. LAB can also survey seed and plant propagules such as endophytes [25]. The carbohydrate-rich environment appears ideal for LAB proliferation. They quickly break down the organic acids and acidify the rhizosphere [29]. At the same time, the acidic environment and weak organic acid exert a toxic effect on other microorganisms. LAB diversity in soils depends on carbon richness, which is abundant in the fruit tree rhizosphere. *Lactobacillus lactis* subsp. *lactis* is broadly distributed in horticultural crops. They have been isolated from the mulberry rhizosphere [27]. Moreover, LAB are halotolerant and survive in low water intensity and high salinity in dry environments. Fhoula et al. (2013) isolated and characterized 119 LAB strains from the rhizosphere of olive trees and desert truffles, and they showed tremendous antimicrobial activity [30].

## 3. Biocontrol Agents of LAB

Fungal contamination of food crops costs the world an estimated USD 60 billion a year in lost agricultural production [31]. About 50% of fruits and vegetables in tropical regions are lost every year due to fungal spoilage [6]. The Food and Agricultural Organization (FAO) estimates that mycotoxin contamination of food crops globally is 25% and could be up to 60–80% [32]. Maize, groundnuts, and tree nuts are the most common foods at risk of contamination with aflatoxins. They are most commonly produced by *Aspergillus*, *Penicillium*, *Fusarium*, and *Alternaria genera*, affecting cereal grains [33]. Among them, *F. oxysporum* is a soil-borne pathogenic fungi that is a significant causative agent in damage to horticultural crops. *Fusarium* wilt is a common disease in the Solanaceae family. *Fusarium* species decrease crop yield and cause considerable losses in banana production. In this context, LAB control pathogenicity in agricultural and horticultural crops, as listed in Table 1. LAB strains are isolated from dairy products and control soil-borne pathogens. In addition, *Lactobacillus buchneri* isolated from corn silages showed antifungal activity against *F. graminearum* [34]. Hamed et al. (2011) demonstrated that seed pre-treatment before planting with an LAB nutritive solution reduces the damping-off diseases [35]. Several studies have shown that LAB could produce antifungal and antibacterial substances to inhibit the growth of pathogenic microbes [7]. Furthermore, lactic acid bacteria, yeast, and phototrophic bacteria culture broth and cell-free extract promote plant growth and protect the plants from abiotic stress [36]. Naturally fermented microbial cocktails are thought to be plant stimulants, and diluted solutions are spraying onto the plant and soil. A simple method to utilize LAB is an aqueous extract/culture filtrated to reduce the *E. coli* population and distribution in fermented food and plants. The earlier implementation of LAB to agricultural and horticultural crops may reduce the risk factors without disturbing the ecosystem. For example, Laury-Shaw et al., demonstrated that an LAB aqueous solution spray could reduce the *E. coli* growth in spinach [37].

## 4. Antibacterial Activity of LAB

LAB strains make different classes of chemical compounds. Among them, the bacteriocins group is the best-studied one. Bacteriocins are toxic to microbes and are the most promising primary metabolites for developing antibiotic drugs. Bacteriocins are peptides or proteins synthesized by ribosomes, and they inhibit the growth and reproduction of a variety of bacteria [55]. Many researchers have proposed the mechanism behind the activity. In addition, bacteriocins may inhibit nucleic acid and protein synthesis [56]. They are divided into two categories. The first is lantibiotics, containing lanthionine or the absence of lanthionine [57,58]. The *Lactobacillus lactis*-derived lanthionine group polycyclic antibacterial peptide causes cell damage in Gram-positive bacteria [59]. The second category of bacteriocins is Helveticin M and Helveticin J, produced by *L. crispatus* and *L. helveticus*. Both bacteriocins are used as food preservatives. Recently, Rooney et al., proposed bacteriocin-mediated resistance in plants to control bacterial pathogens in commercial crops [60].

Furthermore, biosurfactants of bacterial origin have broad applications in the food, agriculture, and pharmaceutical industries. Bacterial origin biosurfactants exhibit antibacterial, antifungal, antimycoplasma, and antiviral properties [12]. Biosurfactants cause membrane damage in pathogens, creating pores on lipid membranes and disrupting porosity and membrane integrity. Additionally, biosurfactants detach microbial cells from surfaces through sloughing, which may cause erosion and abrasion [61]. However, biosurfactants regulate quorum sensing signaling and quorum-sensing-dependent activities. For example, biofilm formation, motility, and pathogenicity are influenced by this signaling. Rodrigues et al., reported that biosurfactants derived from *Lactococcus lactis* inhibit the bacteria and yeast cell adhesion [62]. Fermented dairy products exhibit antimicrobial activity against *E. coli*, while glycolipid biosurfactants responded to the activity [63].

Interestingly, *L. plantarum* significantly reduced the virulence factors and inhibited the biofilm formation of pathogenic bacteria [64]. Lactobacillus rhamnosus effective against *Pseudomonas aeruginosa*, *Staphylococcus aureus* and *E. coli*. Shrestha et al., reported LAB inhibits plant pathogenic bacteria *Ralstonia solanacearum* [65,66]. In addition, *L. plantarum* exhibits antagonistic effects against the phytopathogenic bacteria *P. campestris* [67]. Glycolipid biosurfactants eradicate bacterial biofilm formation and surface adhesion [12]. However, a limited number of strains have been reported for their biosurfactant production ability, antimicrobial potential, and inhibition of biofilm formation. 

## 5. Antifungal Activity of LAB 

Fusarium head blight (FHB) is a severe fungal disease of wheat and cereal crops and affects livestock feed and the quality of seeds. Bafforni et al., demonstrated that *L. plantarum* and *Bacillus* species were applied as biocontrol agents to reduce the FHB index [68]. In addition, LAB increase the nutritional properties of wheat flour and related bakery products and silage. The food-grade LAB can synthesize several promising and eco-friendly metabolites, acting as a biocontrol agent to inhibit molds on fruits and horticultural crops (Table 2). The ascomycete fungus *Zymoseptoria tritici* causes septoria leaf blotch in wheat plants. The primary foliar diseases in wheat are a significant threat to global food grain production. Lynch et al., found that LAB exhibit an antifungal effect against *Z. tritici* [44]. In addition, LAB reduce the toxic agents in wheat and maize grains produced by the filamentous fungi [46,47,53]. De Simone and co-workers demonstrated that the *Lactiplantibacillus plantarum* species exerted strong antagonism against the necrotrophic fungus *Botrytis cinerea* [54]. Grey mold *B. cinerea*, an etiological agent, is a typical contaminant of many horticultural crops. Sathe et al., demonstrated that LAB strains could prolong the shelf life of cucumber [17]. *Lactobacillus plantarum* IMAU10014 exhibits strong antifungal activity against citrus green rot [39]. Crowley and co-workers reported that *Pediococcus pentosaceous* showed a broad spectrum of antifungal activity against fruit crop fungal pathogens [40]. Furthermore, food-grade LAB control the fruit rot diseases caused by *Rhizopus stolonifer* in jackfruit [42]. Matei et al., reported that LAB protect fresh food products against blue mold fungal infection [43]. At the same time, post-harvest decay is the primary source of economic loss, due to infection by the mesophilic fungus *P. digitatum*. *Lactobacillus sucicola* and *Pediococcus acidilactici* showed antifungal activity against *P. digitatum* and other pathogenic species [48]. Several authors reported that LAB exhibits antifungal activity against horticultural and fruit crops [38,49,50,51,54]. On the other hand, Li et al., demonstrated that edible films embedded with 2% LAB prolong shelf life and prevent banana blackening [52]. In addition, the same author observed the antioxidant activity of the composite film, and affirmed its uses in food packaging applications [52].

Nevertheless, increased resistance of pathogenic fungi toward commercial fungicides and climate change impedes the control of fungi in the food supply and necessitates the development of complementary fungicides. LAB-derived metabolites significantly inhibit the pathogenic fungal population and neutralize the mycotoxin levels in fruit and vegetable crops. In addition, they reduce post-harvest decay and inhibit the production of mycotoxins in fermented food products [69]. By increasing the level of the natural antimicrobial compound phenyllactic acid (PLA) during kimchi fermentation, PLA content might enhance the safety of the food products [70]. Furthermore, fatty acids derived from *L. pentosus* exhibit the antifungal activity of various filamentous fungi and yeast pathogens [71]. 3-hydroxyl fatty acid derived from *L. plantarum* inhibited yeasts more actively than filamentous fungi [72]. Lappa et al., demonstrated that LAB act as a potential biocontrol agent against toxigenic fungi in table grapes [73]. In addition, LAB significantly reduced the mycotoxin level in viticulture by 32–92%. LAB combined with carboxymethyl cellulose coatings on fresh strawberries reduced the yeast and mold growth and improved the fruits’ shelf life [49]. In addition, the biocontrol properties of LAB strains on *Cucumis sativus*, *Citrus japonica*, *Selenicereus undatus* (pitahaya), and other fruits and vegetables have also been reported [50]. LAB-derived coriolic acid inhibited the phytopathogenic blast fungi [74]. However, pathogenic fungi are the primary causative agent for fruit deterioration and cause considerable losses in the viticulture industry. The *Lactobacillus plantarum* strain inhibits halos against fungi from *Aspergillus* and *Penicillium* genera. *Lactobacillus plantarum* essential oils combined with a fermented filter showed a synergic antifungal effect against necrotrophic fungus *B. cinerea* [75]. Omedi et al., reported that the phenolic compounds dihydrocaffeic acid, benzoic acid, caffeic acid, phenyllactic acid, p-coumaric acid, and syringic acid showed antifungal activity [50]. LAB strains incorporate an edible coating that protects grapefruits from fungi infection and extends shelf life [76]. However, several authors reported that LAB metabolites showed an antagonistic effect against various economically significant plant pathogenic fungi (Table 2).

**Table 2 ijms-23-07784-t002:** Lactic acid bacteria and their active compounds against plant pathogenic fungi.

Strains	Source	Active Compound	Active Spectrum	References
**Antibacterial**				
*L. plantarum*	Cucumber pickle	Organic acids	*Pseudomonas campestris*	[67]
LAB strain	Tomato rhizosphere	None	*Ralstonia solanacearum*, *Xanthomonas campestris* pv. *vesicatoria*, *Pectobacterium carotovorum* subsp. *carotovorum*	[65,66]
LAB strain	Unknown	None	*Xanthomonas campestris* pv. *vesicatoria*	[65]
*L. lactis*	Curd	Glycolipid biosurfactants	*E. coli*	[63]
**Antifungal**				
*Lactobacillus* species	Type culture	3-Phenyllactic acid	*P. expansum*, *A. flavus*	[13]
*L. acidophilus*	Chicken intestine	Organic acid	*Fusarium* sp., *Alternaria alternate*	[36,77]
*L. amylovorus*	Gluten-free sourdough	Fatty acid, LA, salicyclic acid	*P. paneum*, *Cladosporium* sp., *Rhizopus oryzae*, *Endomyces fibuliger*, *Aspergillus* sp., *Fusarium culmorum*	[36,78,79]
*L. brevis*	Brewing barley	Organic acid, proteinaceous	*A. flavus*, *F. culmorum*, *Trichophyton tonsurans*, *Eurotium repens*, *Penicillium* sp.	[79,80,81].
*L casei*	Dairy products	None	*Trichophyton tonsurans*, *Penicillium* sp.	[80,82]
*L. coryniformis*	Silage, flower, sourdough	PLA, proteinaceous	*Aspergillus* sp., *Fusarium*, *Rhodotorula* sp., *Talaromyces flavus*, *Kluyveromyces* sp.	[77,79]
*L. fermentum*	Fermented food and dairy products	Proteinaceous, PLA	*A. niger*, *Fusarium graminearum*, *A. oryzae*, *A. niger*, *Fusarium* sp.	[83,84]
*L. harbinensis*	Type strain	Fatty acids	*Mucor racemosus*	[85]
*L. lactis*	Wheat semolina	None	*P. expansum*	[82]
*L. mesenteroides*	Raw milk	LA, succinic acid, fatty acids	*Penicillium* species	[86]
*L. plantarum*	Plant materials, food grains, fermented soybean, raw milk	Fatty acids, LA, cyclic dipeptide, phenyllactic acid, peptides, succinic acid	*Broad spectrum*	[53,72,77,86,87,88,89,90,91]
*L. paracasei*	Dairy products, raw milk	Proteinaceous, LA, succinic acid, fatty acids	*Fusarium* sp.	[86,92]
*L. pentosus*	Fruit and fermented food	PLA	*A. oryzae*, *A. niger*, *Fusarium* sp.	[86]
*Pediococcus pentosaceus*	None	Proteinaceous, cyclic acids	*Penicillium* sp., *Aspergillus* sp., *Fusarium* sp., *Rhizopus stolonifer*, *Sclerotium oryzae*, *Rhizoctonia solani*, *Botrytis cinerea*, *Sclerotinia minor*, *Rhodotorula* sp.	[10,17,77,84]
*L. reuteri*	Murine gut, porcine	None	*F. graminearum*, *A. niger*, *Fusarium* sp.	[80,83]
*L. sakei*	Leaves, dandelions, flour	Peptide, PLA	*A. fumigatus*, *Fusarium* species	[77]
*L. salivarius*	Chicken intestine	Peptide, PLA	*A. nidulans*, *F. sporotrichioies*	[77]
*Weissella cibaria*	Food grains, fruits, and vegetables	Organic acids, proteinaceous	*Fusarium culmorum*, *Penicillium* sp., *Aspergillus* sp., *Rhodotorula* sp., *Endomyces fibuliger*	[10,18,93,94]
*W. confuse*	Food grains	Organic acids, proteinaceous	*Penicillium* sp., *Aspergillus nidulans*, *Rhodotorula* sp., *Endomyces fibuliger*	[10,70]
*W. paramesenteroides*	Fermented wax gourd	Organic acids	*Penicillium* sp., *Fusarium graminearum*, *Rhizopus stolonifer*, *Sclerotium oryzae*, *Rhizoctonia solani*, *Botrytis cinerea*, *Sclerotinia minor*	[17,93]

## 6. Biopesticides and Insecticides of LAB

Global climate change and extreme temperatures significantly impact crop production and agricultural pests. Climate change can favor insect and pest populations and prolong their lifespan and survival rate [95]. However, pests and insects cause severe economic damage to many crops and fruit trees. Therefore, the agrochemical industry produces several insecticides and pesticides worldwide. Organophosphorus is a chemical pesticide that causes acute poisoning in humans and animals [96]. Therefore, researchers and the agro-farm industry are looking for alternative tools to prevent agricultural pests. Biopesticides are an alternative to conventional chemical pesticides, and they are eco-friendly and target specific. In addition, microbial-based pesticides comprise numerous microbes such as fungi, bacteria, and nematode-associated bacteria that protect crops from pests and nematodes [97]. For example, LAB species *L. sakei* and *L. curvatus* can efficiently produce metabolites, which tend to kill nematodes [98]. Alawamleh et al., reported that the lactic acid bacteria *Oenococcus oeni* release versatile metabolites and were desirable for spotted wing drosophila [99]. In contrast, the high attraction of fruit fly drosophila results in a high capture rate in traps. However, further study of LAB fermented dairy products in the presence of commercial insecticides that accelerated the acetic condition might have elevated the insecticide activity [100]. Takei et al., demonstrated that LAB enclosing poly(ε-caprolactone) microcapsules are efficient in removing root-knot nematodes [101]. LAB-based microcapsules have been used in horticultural crops to remove root-knot nematodes. In addition, poly(ε-caprolactone) exhibited higher LA production and enhanced the viability and entrapment of LAB cells [101]. In recent years, nanobiotechnology has gained much attention in the agriculture and food sectors. Microbial-based agro-nanotechnology is an eco-friendly approach that might reduce the usage of hazardous chemicals. At the same time, the systematic approach (controlled release) for applying fertilizers and pesticides to crops might enhance the yield and quality of the agro-food. Indeed, nano-based approaches promise an effect on plant health and yield, and these advantages support sustainable agriculture. In addition, nanomaterials have also been tested for pest management of insects in agricultural and urban management [102]. Zinc oxide and silver nanoparticles are widely used due to their antibacterial and antifungal activity [103]. In addition, enzyme-based zinc oxide nanoparticles (ZnONPs) control insect pests and pathogens [104]. The chitinase from *L. coryniformis* immobilizes ZnONPs and its effect on corn lice as a potential insecticide in agricultural bioprocesses, which supports the economy.

## 7. Biostimulants of LAB

Plant-associated microorganisms synthesize phytohormones, and the structure and functional properties are similar. Microbial phytohormones exhibit a similar effect on the plants, and they stimulate or inhibit microbial proliferation [105]. There is limited evidence of LAB-related growth hormones. However, LAB stimulate plant growth and resistance to water and abiotic stress. According to Ampraya et al., LAB exhibit plant-growth-promoting (PGP) properties, and they can produce auxin indole-3-acetic acid (IAA) and solubilize minerals [106]. Lynch precisely reported that the LAB growth hormones cytokinins and other metabolites were found in the soil [107]. In a hypothetical view, LAB gradually incorporated into plant rhizospheric soil may alter plants’ physical properties to maximize the yield. For example, rice seeds coated with *Lactococcus lactis* significantly promoted the root length and shoot length. In addition, *L. lactis* significantly promotes cabbage growth and yield [108]. In addition, several bacterial species produce bacterial exopolysaccharides (EPS) that promote plant growth and enhance soil fertility [105]. LAB-derived EPS exhibits a variety of structural and functional properties. EPS is used in functional food, medicine, and pharmaceuticals. However, there is a lack of evidence on agricultural applications. 

Even though further studies on LAB could enhance organic decomposition and soil humus formation, resulting in high growth and yield in cucumbers [109], high organic matter stimulates specific bacteria populations. It changes the microbiota, which could be highly beneficial to plant and soil fertility. According to a concept formulated by H.P. Rusch, soil fertility of organic agricultural soils can be related to lactic acid bacteria (no literature evidence yet to be disclosed) [110]. Somers et al., found that *Bacillus*, *Paenibacillus*, and *Staphylococcus* species isolated from organic farms significantly promote plant growth in crops [108]. In addition, *Rhodobacter sphaeroides*, *L. plantarum*, and yeast species promote plant growth and increase plant hormones, amino acids, and nutrient content in cucumber [111]. Lutz et al., found that a few *Lactobacillus* strains act as biocontrol and biostimulant agents [112]. LAB colonized in pepper (*Capsicum annum*) rhizosphere produced IAA and siderophore metabolites. LAB strains solubilize phosphate to promote plant growth and control the bacterial spot diseases in pepper [113]. Strafella et al., investigated the comparative genomics and plant growth promotion properties in *L. plantarum* isolated from the wheat rhizosphere [114]. The recombinant *L. plantarum* produced higher succinic acid in the fermented substrate, stimulating plant growth [115]. Several studies have shown that LAB promote plant growth and can act as a biocontrol agent in horticultural crops (Table 3). However, some limitations, often related to plant stimulation effects and inconsistent performance in field conditions, need to promote wide LAB use in agriculture. 

## 8. Biofertilizer of LAB

Anthropogenic ammonia emissions are major risk factors that cause secondary pollution, reduce nitrogen availability, and damage forests and vegetation. Biofertilizers are substances containing a variety of microbes to protect the plant and enhance the plant’s nutrients. However, LAB and nitrification bacteria reduce ammonia emissions and promote nitrification [122]. Recently, LAB and other Bacillus-based biofertilizers have been validated with established microbes in agriculture and the environment. Microbial-based biofertilizers increase crop yield and accelerate the mineral update of the plant root. Further, they enhance the organic matter catabolism (Table 3). Spay and soil injection methods are highly recommended for commercial applications. LAB-based liquid fertilizer spray on the plant and soil is hypothesized to assist plant health. Fermented LAB, yeast, and phototrophic bacteria cocktails are used as biofertilizers and biocontrol agents. At the same time, LAB and bacillus-based biofertilizers showed a high crop yield and enhanced the organic matter degradation (patent no: CA2598539A1, 2006) [123]. In this context, farmyard manure and plant-based compost is integral to organic farming and sustainable agriculture. LAB decompose and bio-stabilize the animal and plant waste to improve the agronomic value and assimilate organic matter such as lignin and cellulose materials [22]. Wang et al., found that *Bacillus stearothermophilus* elevate the relative abundance of LAB strains in soil [122]. In addition, LAB strains exhibit an antagonistic effect against phytopathogenic agents in soil. 

Globally, the mushroom industry has grown rapidly in recent years, with a market value of USD 11.9 million in 2019 [124]. At the same time, spent mushroom substrate (SMS) is a residual material remaining after the harvest: 5 kg of SMS is produced from 1 kg of mushroom harvest. SMS is an alternative animal feed and manure source for horticultural crops [125,126,127]. Compost temperature, average pH, and microaerobic conditions accelerated the LAB growth in SMS. Several LAB species have been isolated from SMS and composting substrate. The most compatible identified in SMS, *L. plantarum*, may have promoted fermentation [128]. Chuang et al., reported that SMS contains multiple constituents, such as a mushroom mycelium, metabolites, organic acid, lactic acid, and polysaccharides. Those metabolites improve animal health and antioxidant capacity while feeding SMS [129].

In addition, LAB-based fermented compost materials increase soil fertility, soil structure, aeration, neutralize alkalinity, and promote moisture retention. Cacace et al., found that LAB produces enormous organic acids during food and backer waste ferment [130]. For this reason, LAB-based composting materials are well suitable for alkaline soils that promote phosphorous and iron precipitates, such as Ca phosphates and iron oxides [131]. Those conditions led to a significant availability of Mn, Fe, and Cu in soils [132]. Some hypothetical views revealed that LAB fix atmospheric nitrogen and produce iron-chelating compounds [130]. However, the comparative genomics data for LAB and food-related strains were differentiated. The recent comparative genomic analysis carried out by Mao et al., provides evidence that the LAB strains differ according to the food niche [133]. Hence, LAB strains exhibit high genomic diversity based on function and substrate, while gene manipulating and metabolic engineering tools alter the gene expression, resulting in plant growth and protection. 

## 9. Soil Bioremediation of Lactic Acid Bacteria

Soil carbon pools are the most significant terrestrial carbon stored in the soil, and they affect the physical, chemical, biological properties. Further, soil organic matter accumulation is crucial for soil fertility, water retention, and crop production. The terrestrial plants utilize inorganic and organic sources of carbon. Hence, modern agricultural practices have negatively impacted the soil ecosystem, due to factors such as intensive tillage, commercial fertilizers, and chemical pesticides. Microorganisms degrade organic and inorganic wastes in soil by the process of bioremediation. Fungi are the predominant species in the soil ecosystem, and they mineralize carbon sources and biosorbent heavy metals from polluted soils. In addition, LAB strains are prevalent in the soil and have been used in the bioremediation process (Figure 3). LAB strains are essential for improving the soil carbon pool, removing heavy metals, and detoxifying the mycotoxins [134,135,136]. Heavy metals are adsorbed by electrostatic and hydrophobic interactions [137]. Therefore, LAB might be used to produce commercial bio-filters to purify water contaminated with heavy metals and aflatoxin. *Lactobacillus plantarum* is a promising biosorbent for removing cationic metals ion such as cadmium and lead from industrial wastewater [135,138]. However, many authors proposed LAB heavy metal biosorption mechanisms [138,139,140], while the bacterial functional groups carboxyl, hydroxyl, and phosphate are involved in this process. Formerly, LAB-based microcapsules exhibited desirable biodegradability properties compared to hydrogel and synthetic polymers [101]. The LA-based microcapsule was more efficient, and the production capacity was comparatively higher than commercial soil amendments. Furthermore, LAB detoxify and degrade pesticides in fermented milk and other fermented food products [141,142]. Zhou and Zhao found that LAB degrade nine different organophosphorus pesticides in dairy products [143].

## 10. Modern Technology and Metabolic Engineering of LAB

LAB degrade macromolecule substances through lactic acid fermentation and produce several metabolic end-products. Hence, LAB metabolites are commercially important, with wide applications in food and medicine [144]. LAB are favorable for metabolism modification since they have a small genome and encode a limited range of biosynthesis capabilities. In recent years, LAB have been receiving much attention as alternative cell factories for the producers of valuable metabolites by metabolic engineering. Genetic manipulation methods have been well established in LAB, promoting industrial application [144]. In addition, LAB strains are widely used in CRISPR-Cas-based genome editing. They are currently a trove of potential for many industries, whether for new vaccine delivery systems or more robust probiotics and starter cultures [145]. The metabolic engineered LAB strains produce lactic acid from an unconventional carbon source, and lactic acid is an essential chemical source for polylactic acid (PLA) and other value-added products. At the same time, metabolic engineered LAB species fermented a considerable quantity of agricultural biomass and produced lactic acid at a low cost with conventional methods. PLA is a biodegradable plastic with excellent biocompatibility and processability. It has been used in agricultural applications such as netting for vegetation and weed prevention [146]. Tsuji et al., demonstrated that recombinant *L. plantarum* produced a higher succinic acid [115]. LAB-derived lactic acid and succinic acids stimulate plant growth. However, the succinic acid fermentation process has not been commercialized yet. Despite these success stories, highly efficient LAB inoculants are not used in sustainable agriculture, although metabolic engineering tools provide efficient cell factories tailor-made to produce beneficial industrial and agro-products. 

## 11. Limitations and Future Prospects of LAB

Since ancient times, lactic acid bacteria have been used as food and medicine, and are the most commonly used probiotics in food. They synthesize various organic acids and other metabolites in the fermentation process. At the same time, the primary acidification process in the fermentation of food and feed substrates prevents the spoilage of microbe populations. Hence, LAB are the most promising candidates for preventing food spoilage and are used as food/feed preservatives [10,75,78,79,129,131]. LAB-derived metabolites are highly beneficial to human and animal health, and are used as food supplements, medicine, and cosmetic products. In contrast, LAB uptake is a high carbon source as an energy source during fermentation, while yielding low biomass and a limited number of metabolites. In addition, acidification and coagulation, low buffering capacity, and sugar depletion are the main limiting factors during fermentation [8,90]. In addition, the high production cost and acidic conditions are drawbacks, limiting the commercial application. However, several studies have pointed out that LAB probiotics are complementary to treating urinary tract infections and respiratory tract infections in humans. However, very limited studies elucidated the role of LAB in the rhizosphere and their plant-growth-promoting properties [27,30]. LAB promote growth in different crops, even though the underlying mechanisms behind this bio-stimulation remain unclear. In addition, LAB showed weak inhibitory activity against plant pathogenic fungi and bacteria. However, LAB exhibited a wide range of antagonistic effects against Gram-positive bacteria. They have minimal effects on Gram-negative bacteria [15]. Plant-growth-promoting properties were limited in LAB, while their performance was poor compared to other beneficial bacteria and fungi. Recently, metabolic engineered microbes have been used in food and agricultural sectors. Although genetically engineered LAB strains positively affect the food and feed industry, fewer studies have investigated agricultural applications. The positive explanation regarding genetically modified LAB was found to have limited evidence, and legal issues limit advanced technology. However, specialization in the LAB gene structure and function and amino acid biosynthesis pathways are warranted. In addition, LAB-based modern farming, LA, PLA, and bacteriocins can be produced sustainably, stimulating technology adoption. LAB strains are highly beneficial to animal health, and they inhibit harmful microbes and promote animal health in nutrition. Various reports have shown that the LAB strains are isolated from forage, control infectious pathogens, and promote the gut microbiota of humans and animals [23,129,147]. In the future, those emerging technologies will increase the yield and build sustainability across crop cultivation and animal husbandry.

## 12. Conclusions

Sustainable agriculture has recently been more concerned with a sustainable food system, and organic farming is most important for global health. Microbial-based agricultural practices would help alleviate these concerns and supply sufficient food for the world population. In this context, novel soil amendments and the exploitation of plant-growth-promoting microorganism potential are promising tools for sustainable agriculture. In contrast, LAB uptake is a high carbon source as an energy source during fermentation with a limited number of yielded metabolites. The acidification and coagulation, low buffering capacity, and sugar depletion of LAB strains are the main limiting factors during mass production. In addition, production cost and high acidic conditions are drawbacks in commercial applications. However, very few studies elucidated the role of LAB and their plant-growth-promoting and biostimulant properties in agricultural applications. In nature, few beneficial microbes that can fit into sustainable agriculture. However, LAB strains are used as a plant growth promoter and biocontrol agent in fruit trees, rice, and horticultural crops. LAB can ferment and decompose animal and mushroom spent substrate waste. They can detoxify the mycotoxin and pesticides in food and feed substrates. In addition, LAB and their antimicrobial and growth-promoting compounds can replace inorganic fertilizer and pesticides. Furthermore, LAB incorporated starch films to protect fruits and vegetables from oxidation damage. This strategy may enhance shelf life without altering the quality of food packaging applications. Recently, LAB encapsulation with different matrices has been used as probiotics in aquaculture [148]. LAB nanomaterials and nano chemicals have appeared as promising agents for plant growth promotion and disease control agents in the near future. The overall agro-based benefits of LAB have been discussed in this review, and we conclude that lactic acid bacteria are a promising candidate for sustainable agriculture.

## Figures and Tables

**Figure 1 ijms-23-07784-f001:**
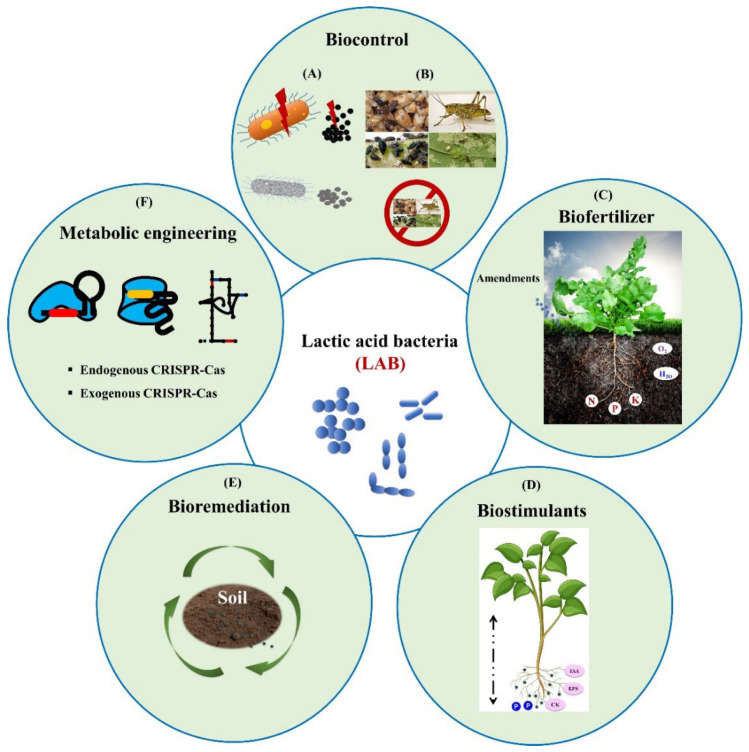
Lactic acid bacteria agricultural application. (**A**). Anti-bacterial and anti-fungal activity; (**B**) biopesticides and insecticides; (**C**) biofertilizer increases soil fertility, aeration and retention of moisture content, elevates the mineral uptake and organic decomposition, acetifies the soil and reduces pest diseases. (**D**) IAA, cytokinin, and siderophore secretion increases the root and shoot length and solubilizes the phosphate in the soil. (**E**) Heavy metal removal, detoxification of fungal mycotoxins, acidification by LA and organic acid, increases organic decomposition, and increases the organic content in the soil, biodegradation. (**F**) CRISPR-Cas systems and derived molecular machines, endogenous or exogenous engineering to enhanced functional attributes.

**Figure 2 ijms-23-07784-f002:**
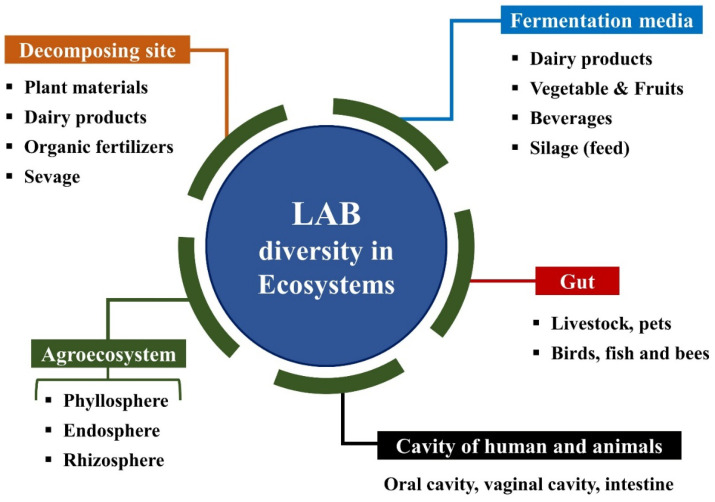
LAB occurrence and dynamism in distinct ecology niches: A widespread application in agricultural, environmental and functional health properties.

**Figure 3 ijms-23-07784-f003:**
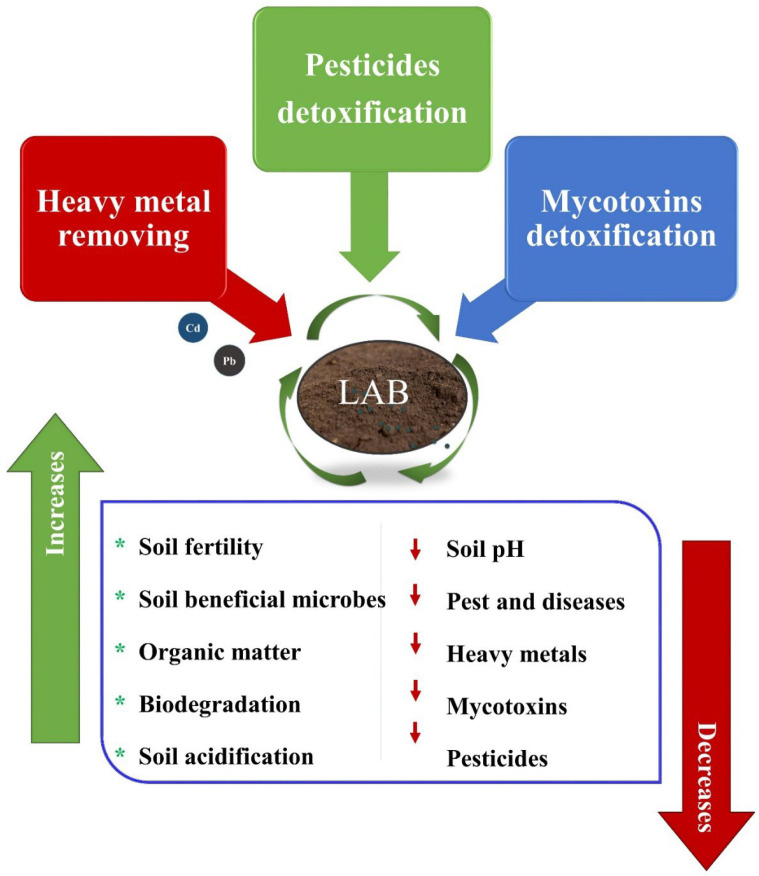
The role of LAB in bioremediation for sustainable agriculture.

**Table 1 ijms-23-07784-t001:** Biocontrol properties of LAB on agricultural and horticultural crops.

Strain Name (LAB)	Pathogens	Food Crops	References
LAB	*Alternaria alternata*	Post-harvest decay	[38]
*Lactobacillus plantarum* CUK-501	*Aspergillus flavu*, *Fusarium graminearum*, *Rhizopus stolonifer*, *B. cinerea*	Cucumber	[17]
LAB	Bacteria and fungi	Vegetables and fruits	[18]
*L. plantarum* IMAU10014,	*Penicillium digitatum*	*Citrus japonica* (kumquat),	[39]
*Pediococcus pentosaceous*	*P. expansum*	*Pyrus* (pear), *Vitis vinifera* (grape), *Prunus* (plum)	[40]
*L. plantarum* LR/14	*A. niger*, *R. stolonifer*, *Mucor racemosus*, *P. chrysogenum*	Wheat seeds	[41]
LAB	*Fusarium*	Cereal-based products	[36]
*Lactococcus lactis* subsp. *lactis*	*Rhizopus stolonifer*	*Artocarpus heterophyllus* (jackfruit)	[42]
Lactic acid bacteria 43, LCM5	*Penicillium expansum*	*Malus domestica* (apple)	[43]
LAB	*Zymoseptoria tritici*	Wheat	[44]
*L. plantarum*	Filamentous fungi and yeast	-	[45]
*Lactobacilli*	*F. verticillioides*	Ensiled corns	[46]
LAB	*Fusarium malting*	Wheat grains	[47]
*L. sucicola*, *P. acidilactici*	*P. digitatum*	Citrus	[48]
*L. plantarum*	-	*Fragaria x ananassa* (strawberry)	[49]
*L. plantarum*, *L. pentosus*, *P. pentosaceus*	*A. niger*, *Cladosporium sphaerospermum*, *P. chrysogenum*	Pitaya (cactus fruit)	[50]
*L. plantarum* TR7	*P. expansum*	*Solanum lycopersicum* (tomato)	[51]
LAB	Blackening	Banana	[52]
*L. plantarum* TE10	*Aspergillus flavus*	Fresh maize seeds	[53]
*L. plantarum*	*Botrytis cinerea*	Horticultural crops	[54]

**Table 3 ijms-23-07784-t003:** LAB biostimulants and biofertilizer properties on sustainable crop production (PGPR—plant-growth-promoting rhizobacteria; IAA—indole acetic acid; LA—lactic acid).

Strains	Source	Crops	Effects	Mechanisms	References
*L. plantarum*	EM-4, type strain, grape must	Radish, tomato	Increased yield, shoot branching, shoot and root growth	None	[35,109]
*L. plantarum*	Grape must, oyster mushroom	Tomato	Increased germination, increased shoot and root growth	Bacteriogenic metabolites	[116]
*L. plantarum*	Commercial phytostimulant	Cucumber	Increased germination and seedling growth	None	[117]
*L. plantarum*	Dairy products	Tomato	Increasing germination rate and root growth	Bacteriogenic metabolites	[116]
*L. plantarum*	Human probiotic	Wheat	Osmotic stress alleviation	None	[118]
*L. plantarum*	PGPR Corp. (Korea)	Cucumber	Increased growth, nutrient uptake, and amino acid content	Increased nutrient availability via succinic acid and LA	[111]
*L. plantarum*	Unknown	Swertia chirayita	Salt stress tolerant	Stress response	[119]
*L. acidophilus*	Dairy products	Tomato	Increased shoot branching, shoot and root growth	None	[35]
*Lactobacillus* sp.	Dairy products	Tomato	Increased shoot branching, shoot and root growth	None	[35]
LAB	Unknown	Pepper	Biocontrol and biostimulant property	IAA and siderophores	[112]
*L. acidophilus*	Wheat rhizosphere	Wheat	Increased plant length and chlorophyll content	IAA	[120]
*L. casei*	Commercial phytostimulant	Cucumber	Increased germination rate	None	[117]
LAB strain KLF01	Tomato rhizosphere	Pepper	Increased root and shoot length, root fresh weight, and chlorophyll content	IAA, phosphate solubilization	[113]
LAB strain KLCO2, KPD03	Unknown	Pepper	Increased root and shoot length, root fresh weight and chlorophyll content	IAA, phosphate solubilization	[113]
LAB strain BL06	Sugarcane ferment	Citrus seedling	Increased height, stem diameter, root and shoot weight	Phosphate solubilization, nitrogen fixation	[121]
LAB	None	None	PGP properties	IAA and mineral solubilization	[106]

## Data Availability

Not applicable.

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
