# Peer review of "Application of Lactic Acid Bacteria (LAB) in Sustainable Agriculture: Advantages and Limitations"

_ijms, 2022, doi:10.3390/ijms23147784_

Round 1

Reviewer 1 Report

In this review article, Raman et el discusses the prons and cons of lactic acid bacteria (LAB) in agro-based products. The authors reviewed existing literature carefully and provided a detailed view on the current advantages as well as limitations. It is well written in most parts and there is some redundancy in the information throughout. This review is of great interest to the researchers in agriculture development.

Minor comments:

1. There are some redundancies at parts about the uses of LAB in agriculture field.

2. Lactic acid bacteria is abbreviated as LAB at many places and Lab in few places which might confuses the readers as Laboratory. This should be corrected.

3. The sentence on line 206 is not clear.

4. In the conclusion, the authors did not mention about the disadvantages associated with LAB in agro-industry. Some of the prevailing disadvantages should be added.

5. There are few grammatical errors which needs to be corrected.

Author Response

Reviewer 1
Comments and Suggestions for Authors
In this review article, Raman et al discuss the pros and cons of lactic acid bacteria (LAB) in agro-based products. The authors reviewed existing literature carefully and provided a detailed view of the current advantages as well as limitations. It is well written in most parts and there is some redundancy in the information throughout. This review is of great interest to researchers in agriculture development.
Minor comments:
1. There are some redundancies in parts about the uses of LAB in the agriculture field. 
Response: We have highlighted the LAB in the agricultural application with proper citation and updated the corrections. 
2. Lactic acid bacteria is abbreviated as LAB in many places and Lab in few places which might confuse the readers as Laboratory. This should be corrected.
Response: We have updated the corrections in the conclusion part, Lab changed to LAB. 
3. The sentence on line 206 is not clear.
Response: We have rewritten the sentence according to the reviewer's suggestion.  
4. In the conclusion, the authors did not mention the disadvantages associated with LAB in the agro-industry. Some of the prevailing disadvantages should be added.
Response: We have included disadvantages and limitations in the conclusion part. 
5. There are a few grammatical errors that need to be corrected.
Response: We have updated the manuscript's grammatical and typo errors, and the English edition has been carried out with MDPI. 

Reviewer 2 Report

This review discusses the benefits of lactic acid bacterial advantages and limitations in sustainable agricultural development. It is important article for human being with possible future clinical application. Lactic acid bacteria (LAB) are more significant groups of probiotic organisms in fermented food and are generally considered safe. As LAB regulates soil organic matters and the biochemical cycle, detoxifies hazardous chemicals, and enhances plant health the review is important for food strategies in the future.

Author Response

This review discusses the benefits of lactic acid bacterial advantages and limitations in sustainable agricultural development. It is an important article for human beings with possible future clinical applications. Lactic acid bacteria (LAB) are more significant groups of probiotic organisms in fermented food and are generally considered safe. As LAB regulates soil organic matters and the biochemical cycle, detoxifies hazardous chemicals, and enhances plant health the review is important for food strategies in the future.
Response: Thank you for your review, we appreciate your feedback. 
